# Comparing Short *Versus* Long Persistency of Anthelmintics: Impacts on Dairy Sheep Production

**DOI:** 10.3390/ani15071028

**Published:** 2025-04-02

**Authors:** Konstantinos V. Arsenopoulos, Eleni Michalopoulou, Elias Papadopoulos

**Affiliations:** 1Department of Veterinary Medicine, School of Veterinary Medicine, University of Nicosia, 2414 Nicosia, Cyprus; 2Institute of Veterinary Pathology, Vetsuisse Faculty, University of Zurich, 8057 Zurich, Switzerland; echmicha@gmail.com; 3Laboratory of Parasitology and Parasitic Diseases, School of Veterinary Medicine, Faculty of Health Sciences, Aristotle University of Thessaloniki, 54124 Thessaloniki, Greece; eliaspap@vet.auth.gr

**Keywords:** eprinomectin, albendazole, milk yield, fat and protein concentration, somatic cell count, dairy ewes

## Abstract

This study compared the short-term persistent efficacy of albendazole and the long-term persistent efficacy of eprinomectin in treating gastrointestinal nematodes in dairy ewes and assessed their impact on milk yield and quality. Ewes were divided into four groups, including a control group and those treated with albendazole and pour-on or injectable eprinomectin. Fecal egg counts, milk yield and milk composition were monitored over 75 days. Results showed that eprinomectin, both pour-on and injectable, provided prolonged protection, significantly reducing parasite loads and improving milk yield, fat and protein content and somatic cell counts. These findings support eprinomectin as a superior anthelmintic choice for lactating ewes, enhancing productivity and milk quality while mitigating parasite resistance.

## 1. Introduction

In Greece, sheep farming is mainly focused on milk production, with the majority of Greek dairy sheep reared under semi-intensive management conditions. In these systems, feeding regimes are generally characterized by a combination of grazing on natural pastures year-round and supplementary feeding of concentrates and forages (i.e., alfalfa hay) during winter [1]. Under these circumstances, grazing dairy ewes are often heavily challenged by parasitic infections, which represent one of the most important concerns in sheep farming due to their negative consequences on milk and meat yield, lamb growth and ewe fertility. These consequences are associated with increased culling, an increased replacement rate and predisposition to other diseases, leading to significant economic losses [2,3,4,5,6].

Nematode genera such as *Teladorsagia*, *Haemonchus*, *Trichostrongylus*, *Nematodirus* and *Chabertia* are representative of the wide spectrum of gastrointestinal nematodes (GIN) infecting sheep throughout Europe [6]. These GIN have been detected with variable prevalence among sheep of central and northern Europe, e.g., Austria [7], the United Kingdom [8] and Norway [9], as well as among Mediterranean European countries, e.g., Spain [10], Italy [11,12] and Greece [13,14].

Control of GIN infections in dairy ewes is demanding and is based on timely anthelmintic treatments [15,16]. Despite the wide variety of commercially available anthelmintic drugs (e.g., albendazole) against ovine GIN, their use has certain limitations, with the withdrawal periods during lactation being the most important among them [5,6,17]. For example, albendazole is an anthelmintic drug, which requires a 4-day withdrawal period, and, therefore, it is not recommended for lactating ewes. Furthermore, the development of benzimidazole-resistant GIN places another restriction on the effectiveness of parasite control strategies, which needs to be taken under consideration [16,18].

A relatively novel macrocyclic lactone, eprinomectin is registered for use in lactating ewes to overcome these limitations. Eprinomectin combines a wide spectrum antiparasitic activities against GIN, lungworms and some ectoparasites [11] and a zero-day milk withdrawal period [19]. Eprinomectin is commercially available in Greece as a pour-on formulation for use in dairy cattle and small ruminants at doses of 0.5 mg/kg and 1.0 mg/kg [20], respectively, and as injectable solution registered for dairy cattle and small ruminants at a dose of 0.2 mg/kg [21].

The anthelmintic efficacy of eprinomectin has been studied either as a pour-on formulation in sheep [6,22,23], goats [24] and cattle [25] or as an injectable one in sheep [26,27], goats [28] and cattle [28,29]. However, there is no research comparing the treatment of GIN-infected dairy sheep with eprinomectin (i.e., anthelmintic drug with long-persistent efficacy) and albendazole (i.e., anthelmintic drug with short-persistent efficacy), plus their effects on both the quantitative and qualitative parameters of their produced milk. Therefore, the objectives of the present study were (i) to compare the short (i.e., albendazole) *versus* long (i.e., eprinomectin) persistent effects of these anthelmintics and (ii) to determine these effects’ impacts on milk yield and milk quality in dairy ewes of the Chios breed naturally infected with GIN.

## 2. Materials and Methods

### 2.1. Flocks’ History

Twelve farms from Thessaly, Central Macedonia, Epirus and Thrace, mainland Greece (i.e., three farms per region), were included in this study. The average flock size was ca. 250 ewes with an average milk yield of 260 L per ewe per lactation. The duration of the lactation period was 240 days. All included farms were following the principles of the semi-intensive management system, which is typical for Greece [30]. Their feeding comprised mainly grazing on natural pastures for at least 6 to 8 h per day throughout the study period. A mixed concentrate feed, up to 1.0 kg, was provided twice per day in the milking parlor. The animals also received a fixed amount of alfalfa hay (up to 1.0 kg) and had *ad libitum* access to wheat straw and water.

The vaccination and antiparasitic programs were similar for the participating flocks. The ewes were vaccinated against *Clostridium* spp. (Convexin 8A^®^, Pfizer, Wellington, New Zealand) 20 days before parturition and six months later. Another vaccination against *Mycoplasma agalactiae* (Agalax^®^, SYVA, Leon, Spain) was performed 30 days before parturition and 6 months later. The antiparasitic program included *per os* benzimidazole drugs (10 mg/kg BW) once at lambing. Experimental ewes had not received any anthelmintic treatment in the past 5 months before the start of this study.

### 2.2. Experimental Design

This study was conducted from April to July 2024. Four hundred and eighty clinically healthy adult dairy ewes, from the 2nd to the 4th lactation period, in good body condition score (i.e., between 2.5 and 3.5 on the five-grade scale proposed by Russel et al. [31]) were selected based on their fecal egg counts (FECs) and included in this study. All ewes were milked twice per day and were at the middle of lactation (i.e., 4 months post-lambing).

On each farm, forty ewes expelling more than 300 GIN eggs per gram of feces (epg) were selected and randomly divided into four similar groups (Group 1, 2, 3 and 4). More precisely, ewes of Group 1 (n = 10) remained untreated (control group), those of Group 2 (n = 10) were treated with albendazole *per os* (Albendazole^®^ 10%, Provet, Kharkin, Ukraine) at a single dose of 5.0 mg/kg BW, those of Group 3 (n = 10) were administered eprinomectin pour-on (Eprinex Multi^®^ 5 mg/mL, Boehringer Ingelheim, Dortmund, Germany) at a single dose of 1.0 mg/kg BW and the animals of Group 4 (n = 10) were administered injectable eprinomectin (Eprecis^®^ 20 mg/mL, Ceva Animal Health, Libourne, France) at a single dose of 0.2 mg/kg BW.

### 2.3. Fecal Sampling and Parasitological Procedures

Individual fecal samples were collected from each ewe of all groups on Days 0, 15, 30, 45, 60 and 75. All samples were collected directly from the rectum of each ewe, using sterile plastic gloves and lubricant. After their collection, they were stored in an isothermal container (+2 °C to +4 °C) before transfer to the Laboratory of Parasitology and Parasitic Diseases, the Faculty of Health Sciences of the Aristotle University of Thessaloniki. Sampling was carried out in accordance with animal welfare and did not cause any significant stress to the animals.

The quantitatively modified McMaster technique, expressing a sensitivity of 50 epg, was performed to assess the individual FECs for nematode parasite eggs [32]. Furthermore, pooled fecal samples, from each farm, were processed for coprocultures on the forementioned sampling occasions. The Baermann technique was used for the nematode larvae recovery after an incubation period of 7–10 days [33]. One hundred (when possible) parasitic nematode L_3_ larvae were morphologically identified, following the morphological keys of Van Wyk and Mayhew [34].

### 2.4. Milk Sampling for the Determination of the Fat and Protein Concentrations and Somatic Cell Counts

Individual milk samples (ca. 50 mL), containing a representative volume of milk from each udder half, were collected from each ewe on Days 0, 15, 30, 45, 60 and 75. Sheep were walked calmly to the milking parlor, where they were individually restrained. Teat ends were sanitized using a cotton ball soaked in 90% isopropyl alcohol. After the withdrawal of the initial 2–3 squirts of milk, a milk sample were taken in sterilized plastic tubes with lids. After milk sample collection, teats were disinfected with a solution of 1% iodine to prevent the transmittance of bacteria into the teat canal. After their collection, all tubes were stored in an isothermal container (+2 °C to +4 °C) before transfer to the Laboratory of a private Greek dairy industry testing facility. The determination of the fat and protein concentrations was performed by infrared analysis (FTIR interferometer) using an automatic high-throughput analyzer MilkoScan™ FT6000 (Foss Electric, Hillerød, Denmark). Somatic cell count (SCC) was evaluated by flow cytometry using Fossomatic™ FS (Foss Electric, Hillerød, Denmark).

### 2.5. Determination of the Daily Milk Yield

After the previous described milk sampling procedure, ewes were hand-milked and the volume of the produced milk per ewe per milking was recorded using a graduated measuring cylinder (Ilmenau Company, Ilmenau, Germany). Daily milk yield was estimated using one of the two milkings (morning or evening) and by making adjustments according to ICAR recommendations (AT4 method) [35].

### 2.6. Statistical Analysis

Data were analyzed using STATA 13 (StataCorp. 2013. Stata: Release 13. Statistical Software). The assumptions of the normality and homogeneity of variances for the continuous variables were tested using the Shapiro–Wilk test and Levene’s test, respectively. Non-parametric methods (Wilcoxon rank-sum/Mann—Whitney test) were used during univariate analysis since none of the variables were normally distributed or possible to transform to normal distribution.

Generalized Estimating Equations (GEEs) were used to model the associations of FECs, milk yield, milk fat and protein concentrations and SCC with the different treatments and sampling dates. Robust standard errors were used in the models to address clustering around individual animals for different sampling dates. Model selection was carried out using quasi-likelihood under the independence model criterion (QIC) [36].

The overall efficacy of each antiparasitic treatment at each group was calculated as follows:$$\left(\%\right)\;\mathrm{efficacy}=100\times \frac{C-T}{C},$$
where C represents the geometric mean FEC for control ewes (Group 1) and T the geometric mean FEC for anthelmintic-treated ewes (Group 2, 3 and 4). The differences in the proportions of L3 larvae (per parasite taxon) per sampling occasion among four groups of ewes were evaluated with the chi-squared proportion test. Statistical significance was set at *p* ≤ 0.05.

## 3. Results

### 3.1. Prevalence of Gastrointestinal Nematodes Identified

The most prevalent nematode parasites, found in the studied ewe population from all areas on Day 0, were *Teladorsagia* spp., *Haemonchus* spp. and *Trichostrongylus* spp. at percentages of 52%, 41% and 3%, respectively. Proportionally, *Teladorsagia* spp., along with *Haemonchus* spp., remained the same across the study period in Groups 1, 3 and 4. In Group 2, a significant change in the proportions of these two most prevalent GIN was recorded (Table 1). No other helminths (i.e., trematodes, cestodes) were detected.

### 3.2. Determination of the Efficacy of the Anthelmintic Drugs

Overall, the efficacy of the long-persistent anthelmintic was over 92% for the 75-day period of the present trial. On the contrary, the efficacy of the short-persistent anthelmintic was lower compared to the long-persistent one throughout the study period, reaching up to 90% (Table 2).

### 3.3. Effects of the Anthelmintic Treatment on Fecal Egg Counts in the Ewe Population

As presented in Table 3, untreated ewes had significantly higher (*p* ≤ 0.001) parasitic loads (epg) in comparison to the ewes treated either by injectable and pour-on eprinomectin or albendazole (decreased by ca. 1248 epg (*p* ≤ 0.001, 95% CI, −1349 to −1146), 1270 epg (*p* ≤ 0.001, 95% CI, −1376 to −1165) and 1048 epg (*p* ≤ 0.001, 95% CI, −1141 to −955), respectively). Furthermore, the FECs (epg) of Group 2 were significantly higher (*p* ≤ 0.05) than those of Groups 3 and 4.

The evolution of the FECs (epg) across the studied period for the four different treated groups is presented in Figure 1. The FECs (epg) of the control group (i.e., Group 1) were significantly higher (*p* ≤ 0.001) than those of Groups 2, 3 and 4 in every sampling occasion after Day 0. Moreover, the FECs (epg) between eprinomectin-treated Groups 3 and 4 did not present any statistical differences (*p* > 0.05) on every sampling occasion after Day 0.

### 3.4. Effects of the Anthelmintic Treatment on Daily Milk Yield in the Ewe Population

As presented in Table 4, injectable (increased by ca. 241 mL (*p* ≤ 0.001, 95% CI, 168 to 315)) and pour-on (increased by ca. 168 mL (*p* = 0.05, 95% CI, 78 to 244)) eprinomectin-treated ewes produced significantly higher daily milk yields (mL) in comparison to the untreated ewes. Furthermore, the daily milk yields (mL) of albendazole-treated ewes were significantly lower (*p* ≤ 0.05) than the respective one of the injectable eprinomectin-treated ewes.

The evolution of daily milk yield (mL) across the studied period for the four different treated groups is presented in Figure 2. The daily milk yield (mL) of the untreated and albendazole-treated ewes was significantly lower (*p* ≤ 0.05) when compared to the daily milk yield (mL) of injectable eprinomectin-treated ewes on every sampling occasion after Day 0.

### 3.5. Effects of the Anthelmintic Treatment on Milk Fat Concentration in the Ewe Population

As presented in Table 5, ewes of the control group (i.e., Group 1) produced milk with significantly (*p* ≤ 0.001) lower fat concentration (%) when compared with the milk produced by those of Groups 2, 3 and 4 (increased by ca. 0.193% (*p* ≤ 0.001, 95% CI, 0.148 to 0.238), 0.293% (*p* ≤ 0.001, 95% CI, 0.250 to 0.336) and 0.465% (*p* ≤ 0.001, 95% CI, 0.405 to 0.526), respectively). Furthermore, ewes of Group 3 and 4 produced milk with higher (*p* ≤ 0.05) fat concentrations (%) when compared to those of Group 2.

The evolution of the milk fat concentration (%) across the studied period for the four different treated groups is presented in Figure 3. The milk fat concentration (%) of the control group (i.e., Group 1) was significantly lower (*p* ≤ 0.05) than those of Groups 2, 3 and 4 on every sampling occasion after Day 0. Moreover, the milk fat concentrations (%) of Group 2 were significantly lower (*p* ≤ 0.05) than those of Groups 3 and 4 on every sampling occasion after Day 0.

### 3.6. Effects of the Anthelmintic Treatment on Milk Protein Concentration in the Ewe Population

As presented in Table 6, ewes of the control group (i.e., Group 1) produced milk with a significantly (*p* ≤ 0.001) lower protein concentration (%) when compared with the milk produced by those of Groups 2, 3 and 4 (increased by ca. 0.075% (*p* ≤ 0.001, 95% CI, 0.046 to 0.104), 0.191% (*p* ≤ 0.001, 95% CI, 0.135 to 0.248) and 0.247% (*p* ≤ 0.001, 95% CI, 0.185 to 0.309), respectively). Furthermore, ewes of Group 3 and 4 produced milk with a higher (*p* ≤ 0.05) protein concentration (%) when compared to those of Group 2.

The evolution of milk protein concentration (%) across the studied period for the four different treated groups is presented in Figure 4. The milk protein concentration (%) of Group 1 was significantly lower (*p* ≤ 0.05) than those of Group 2, 3 and 4 on every sampling occasion after Day 0. Moreover, the milk protein concentration (%) of Group 2 was significantly lower (*p* ≤ 0.05) than the respective ones of Groups 3 and 4 on every sampling occasion after Day 15.

### 3.7. Effects of the Anthelmintic Treatment on Somatic Cell Count in the Ewe Population

Finally, as presented in Table 7, in Groups 2, 3 and 4, SCCs were significantly reduced by ca. 128,324 (*p* ≤ 0.001, 95% CI, −186,188 to −70,460), 265,724 (*p* ≤ 0.001, 95% CI, −323,246 to −208,202) and 199,322 (*p* ≤ 0.001, 95% CI, −263,347 to −135,297) cells/mL, respectively, compared to those of the control group (i.e., Group 1).

Figure 5 presents the SCC evolution of the four studied groups during the study period. The SCCs of the untreated ewes were significantly higher (*p* ≤ 0.001) when compared with those of Groups 2, 3 and 4 on every sampling occasion after Day 0.

## 4. Discussion

To the best of our knowledge, this is the first field study comparing the short persistency of albendazole *versus* the long persistency of eprinomectin and their effects on milk yield and quality (i.e., fat and protein concentration and SCC) in twelve semi-intensive dairy sheep flocks of the Chios breed naturally infected with GIN.

Albendazole is a benzimidazole registered to be used as an oral formulation (i.e., tablets or liquid solution) with a short-persistent efficacy against nematode parasites of the gastrointestinal and respiratory system at a dose of 5 mg/kg BW. It is also effective against tapeworms and adult liver flukes at higher doses [37]. Even though albendazole is widely used in most antiparasitic regimes, it is accompanied by some severe restrictions. The first one is its prohibition for use during the lactation period due to the milk withdrawal period. More specifically, albendazole requires a four-day withdrawal period in milk, and, therefore, it is not recommended for lactating ewes. Instead, it can be administered only in ewes during the dry period or post-lambing [26]. Secondly, the irrational use of this short-persistent anthelmintic drug in most anthelmintic regimes has enhanced the development of benzimidazole resistant GIN strains, a threat for the sheep farming worldwide. Parasite resistance to benzimidazoles has been associated with the repeated use of this group of drug in many countries, which, in turn, has reduced both their effectiveness and use [37]. In Greece, benzimidazole-resistant nematode strains, particularly *Teladorsagia* spp. and *Haemonchus* spp., have been reported [18,38]. In the current study, the efficacy of benzimidazoles was lower than 90% throughout the study period. This low efficacy of benzimidazoles, combined with the altered proportion of the identified parasitic genera post-albendazole treatment (Group 2), further re-confirmed the development of resistance against benzimidazoles [39].

Eprinomectin, the newest developed member of the avermectin sub-group of macrocyclic lactones, has been licensed for use to overcome the aforementioned restrictions. It has a broad anthelmintic spectrum against gastrointestinal nematodes, lungworms and some ectoparasites, and it is characterized by a long persistency of 28 days [11,40]. It can be included in the anthelmintic regimes of ewes during the lactation period, exhibiting a broad safety margin and minimal milk elimination. Currently, this long-persistent anthelmintic is available either as a pour-on or injectable formulation for use in lactating dairy cattle and small ruminants with a zero-day milk withdrawal period. In the current study, eprinomectin, administered in both forms, was highly effective against GIN (i.e., 92.1–99.9% efficacy), and the proportion of the identified parasitic genera remained stable post-eprinomectin treatment throughout the study. These two results may indicate the absence of species-specific resistance against eprinomectin until now. Many studies conducted in cattle [25], sheep [6,22] and goats [24] have confirmed the high anthelmintic efficacy of eprinomectin administered topically or by injection at the recommended dosage for each ruminant species. Another noteworthy result was that both forms of this long-persistent anthelmintic drug exhibited an excellent prolonged anthelmintic efficacy up to 10 weeks post-treatment. This finding has been confirmed by previous studies, which recorded a prolonged eprinomectin efficacy against experimental or natural GIN-infected sheep, goats and cattle for a period over 28 days post-treatment [27,41,42].

In general, the effects of the different antiparasitic regimes on the milk yield of dairy ewes has been widely reported, with the majority of the scientific community agreeing with their beneficial outcomes on daily milk yield [27,43,44,45,46]. Indeed, the beneficial effect of albendazole administration via the oral route at a dose of 3.8 mg/kg BW on the daily milk yield of dairy ewes has been previously confirmed [43,46]. However, our results did not demonstrate a significant effect of this short-persistent anthelmintic on the daily milk yields of dairy ewes, even at a dose of 5 mg/kg BW. We anticipated a reduced efficacy of pro/benzimidazoles against GIN and, therefore, no positive effect on daily milk yield, since our previous field studies indicated that the extensive use of pro/benzimidazoles has enhanced the development of benzimidazole-resistant GIN, posing an emerging threat for the Greek sheep farming sector [18,38].

On the contrary, eprinomectin treatment significantly increased the produced milk volume, ranging from 78 to 315 mL per animal per day, compared to the control group of ewes in our study. This effect agrees with previous studies conducted on grazing dairy ewes in Greece. An increase in the produced milk from 50 to 100 mL per animal per day was recorded for dairy ewes treated once with pour-on [23] and injectable [27] eprinomectin, respectively, compared to the untreated control ewes. This increase in milk production observed in our study is attributed to the greater nutrient supply provided to the treated ewes during the long-persistent period post-eprinomectin administration, which was sufficient to enhance milk production [45]. It must be mentioned that the highest increase in milk production was associated with the injectable form of this long-persistent anthelmintic drug (i.e., ca. 241 mL more milk per ewe per day). According to Lespine et al. [47], the subcutaneous administration of eprinomectin presented 2.5 times higher bioavailability than the pour-on application in dairy goats. Another novel finding of our study was that ewes treated with the short-persistent albendazole produced 108 to 181 mL less milk per animal per day compared to those treated with the long-persistent one, suggesting a superior efficacy of eprinomectin in enhancing daily milk yield. One potential explanation for the observed difference is that eprinomectin can provide a more prolonged and effective reduction in parasitic burdens compared to albendazole. This enhanced parasitic control likely decreases physiological stress and inflammatory response in the ewes, thereby allowing more nutrients to be allocated toward milk production rather than to combating infection [45].

Mediterranean countries, including Greece, produce about one third of the world’s sheep and goat milk, which is mainly used for the production of cheese and yogurt. It is obvious that the improvement of milk quality is desirable not only for the farmers, as it determines the final price of the produced milk, but also for the milk industry, as it affects the production of cheese and yogurt [48]. In addition, the quality of the produced milk is a significant indicator of the health status of a ewe. For example, a decrease in the milk fat concentration is associated with low production of acetic acid in the rumen, which, in turn, is linked to the low fiber content of the feedstuff and increased risk of ruminal acidosis [49]. In this direction, factors such as the parasitic burden, which may affect the quality of the milk, should be taken into consideration; thus, it is important to measure the effects of GIN parasitism on milk composition [48], despite their controversial results in the literature. According to Rinaldi et al. [50], goats infected with GIN demonstrated 29.9% and 23.3% lower milk fat and protein contents, respectively, in their milk samples than the control ones. On the contrary, Cruz-Rojo et al. [48] did not record any significant variations in milk fat concentration between high- and low- *Teladorsagia circumcincta* infections of dairy ewes of the Spanish Assaf breed.

In our study, GIN-infected ewes (i.e., untreated ewes) produced milk with significantly lower fat (i.e., 19.3–46.5% less fat content) and protein (i.e., 7.5–24.7% less protein content) concentrations throughout the study period compared to the milk produced by anthelmintic-treated ewes. To be specific, our statistical analysis revealed that every 1% increase in the milk fat content was associated with a ca. 0.00034 epg lower GIN infection (*p* < 0.001). Trichostrongylidae infections in small ruminants disrupt nutrient digestibility and absorption, which can lead to a decrease in voluntary feed intake [51,52]. Since milk fat is synthesized from fatty acids originating from both the animal’s body reserves and dietary intake, compromised nutrient uptake not only diminishes body condition but also hinders the absorption of dietary fats, particularly long-chain fatty acids, thereby further reducing milk fat content [51,52]. The respective association between milk protein (%) content and FEC (epg) was not significant (*p* = 0.140), according to our statistical analysis of the data provided. However, to our surprise, we recorded that every extra 1% of milk protein produced was associated with ca. 0.15% more milk fat content (*p* < 0.001). This indicates that the reduced protein content of the milk produced by the GIN-infected ewes was not directly associated with the GIN burden in terms of the FEC (epg) but with other predisposing factors such as energy availability. Indeed, we know that the reduced digestibility and absorption of dietary fat, as described above, influences casein synthesis, contributing to the low protein content of milk [53].

Several studies in the literature evaluated the effect of the short-persistent anthelmintic (i.e., albendazole) on milk components [43,46], even though there is still a lack of research regarding the effect of the long-persistent anthelmintic (i.e., eprinomectin) on the qualitative parameters of milk produced by dairy ewes. No effects of eprinomectin administration on the milk fat and protein concentrations of dairy ewes were reported by Termatzidou et al. [27]. Studies conducted in grazing dairy cows reported that variations in milk components were similar between pour-on eprinomectin-treated animals and controls [54,55,56,57]. However, McPherson et al. [58] recorded higher milk fat concentration and a tendency for higher milk protein concentration in multiparous cows treated with the long-persistent anthelmintic topically. According to our findings, both short- and long-persistent anthelmintic treatments improved milk quality compared to controls, but the latter one was notably more effective. This enhanced performance may be due to its longer-lasting action and superior ability to reduce the parasitic load, along with the absence of eprinomectin-resistant GIN compared to albendazole, resulting in better nutrient availability and allocation for synthesizing milk fat and protein [43].

Finally, another interesting point of our study was the significant reduction in the SCC in the produced milk of the experimental ewes treated either with short- or long-persistent anthelmintic drugs compared to control ewes, improving not only the health status of the udder but also the milk quality. Our statistical analysis revealed that for every 1 epg increase in the FEC, ewes produced milk with ca. 92.71 more somatic cells (*p* < 0.001). This finding further confirmed a previous study conducted by Arsenopoulos et al. [23] in Greece. Studies using the long-persistent anthelmintic in dairy cows supported the positive relationship between pour-on eprinomectin administration and a reduction in the milk SCC, in agreement with our results. More precisely, Reist et al. [54,57] and Zaffaroni et al. [59] reported a reduction in the SCCs of grazing cows after the administration of the aforementioned antiparasitic drug in these animals. In contrast to our results, Cruz-Rojo et al. [48] found that the administration of the short-persistent anthelmintic to Spanish Assaf dairy ewes had no effect on SCC across the experimental period, apart from on Days 70 and 126 after lambing, where the ewes highly infected with *T. circumcincta* presented their highest number of somatic cell counts. At the same time, Termatzidou et al. [27] did not record a significant effect of the injectable eprinomectin administration on the reduction in milk SCC. It is well known that GIN parasitism, especially parasite genera of *Teladorsagia* and *Haemonchus,* can lead to protein and antibody deficiency [60,61]. This reduction in antibodies suppresses the defense mechanisms of the mammary gland of the ewe, leading to increased SCC and the appearance of clinical [62] or subclinical mastitis [63]. In addition, GIN reduces energy availability (i.e., parasites engage energy sources), predisposing ewes to decreased neutrophil function [64] and, therefore, to mastitis [62]. Implementing effective parasite control with long-persistent anthelmintics, alongside enhanced nutritional supplementation to boost protein and energy intake and rigorous udder health monitoring with early intervention strategies, constitutes a comprehensive approach to mitigate the immunosuppressive effects of GIN parasitism and reduce the risk of mastitis in dairy ewes.

## 5. Conclusions

This study demonstrated the superior efficacy of the long persistent eprinomectin compared to the short-persistent albendazole in controlling GIN infections in dairy ewes. Both pour-on and injectable eprinomectin formulations provided long-lasting protection, significantly reducing fecal egg counts for an extended period. Additionally, ewes treated with the long persistent anthelmintic drug exhibited increased daily milk yield and improved milk composition in terms of fat and protein concentrations. The reduction in the SCCs in treated groups further indicated improved udder health and overall milk quality. These findings highlight the benefits of the long persistent efficacy of eprinomectin as a preferred anthelmintic treatment for lactating dairy ewes, offering enhanced productivity and milk quality while addressing parasite resistance concerns.

## Figures and Tables

**Figure 1 animals-15-01028-f001:**
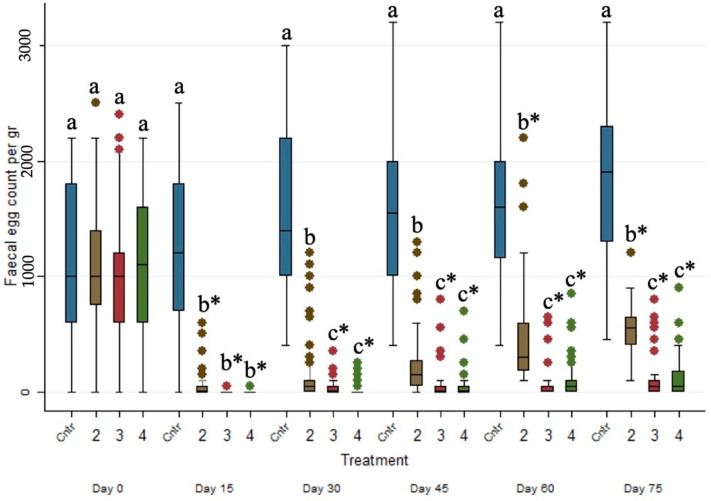
Box plots indicating the effects of the different anthelmintic treatments on fecal egg count (epg) among 4 different groups per sampling occasion —Group 1: control group or untreated animals (blue color); Group 2: albendazole-treated animals (brown color); Group 3: pour-on eprinomectin-treated animals (red color); Group 4: injectable eprinomectin-treated animals (green color). a, b and c indicate statistically significant differences (*p* ≤ 0.05) among the 4 different treated groups per sampling; * indicates statistical differences at *p* < 0.001.

**Figure 2 animals-15-01028-f002:**
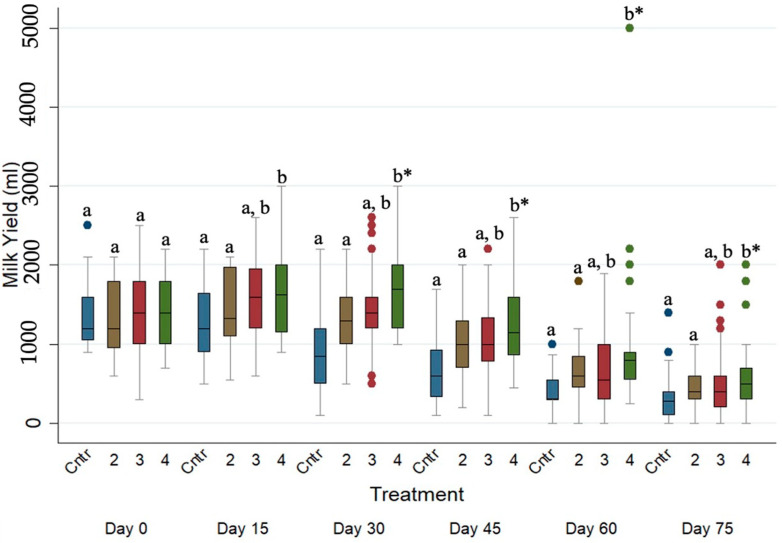
Box plots indicating the effects of the different anthelmintic treatments on daily milk yield (mL) among 4 different groups per sampling occasion —Group 1: control group or untreated animals (blue color); Group 2: albendazole-treated animals (brown color); Group 3: pour-on eprinomectin-treated animals (red color); Group 4: injectable eprinomectin-treated animals (green color). a and b indicate statistically significant differences (*p* ≤ 0.05) among 4 different treated groups per sampling; * indicates statistical differences at *p* < 0.001.

**Figure 3 animals-15-01028-f003:**
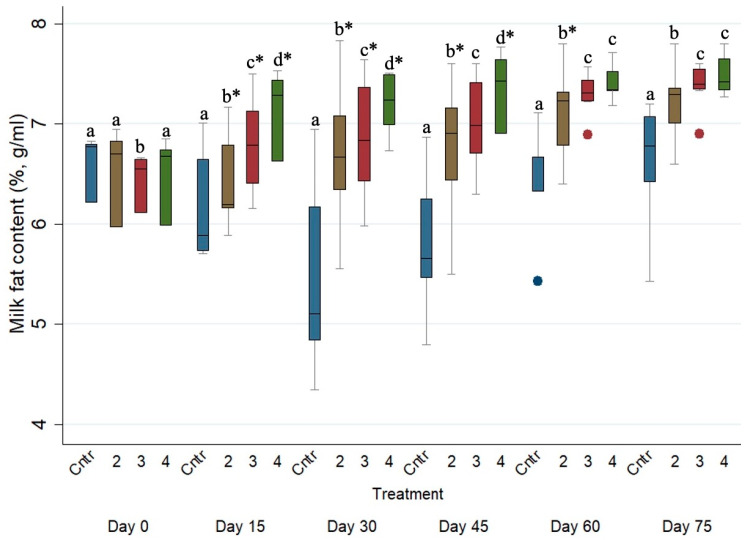
Box plots indicating the effects of the different anthelmintic treatments on milk fat concentration (%) among 4 different groups per sampling occasion—Group 1: control group or untreated animals (blue color); Group 2: albendazole-treated animals (brown color); Group 3: pour-on eprinomectin-treated animals (red color); Group 4: injectable eprinomectin-treated animals (green color) per sampling occasion. a, b, c and d indicate statistically significant differences (*p* ≤ 0.05) among the 4 different treated groups per sampling; * indicates statistical differences at *p* < 0.001.

**Figure 4 animals-15-01028-f004:**
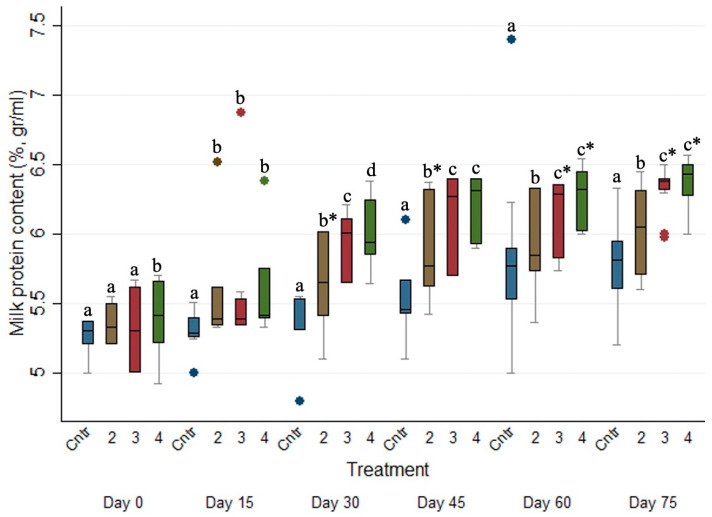
Box plots indicating the effects of the different anthelmintic treatments on milk protein concentration (%) among 4 different groups per sampling occasion—Group 1: control group or untreated animals (blue color); Group 2: albendazole treated animals (brown color); Group 3: pour-on eprinomectin treated animals (red color); Group 4: injectable eprinomectin treated animals (green color). a, b and c indicate statistically significant differences (*p* ≤ 0.05) among the 4 different treated groups per sampling; * indicates statistical differences at *p* < 0.001.

**Figure 5 animals-15-01028-f005:**
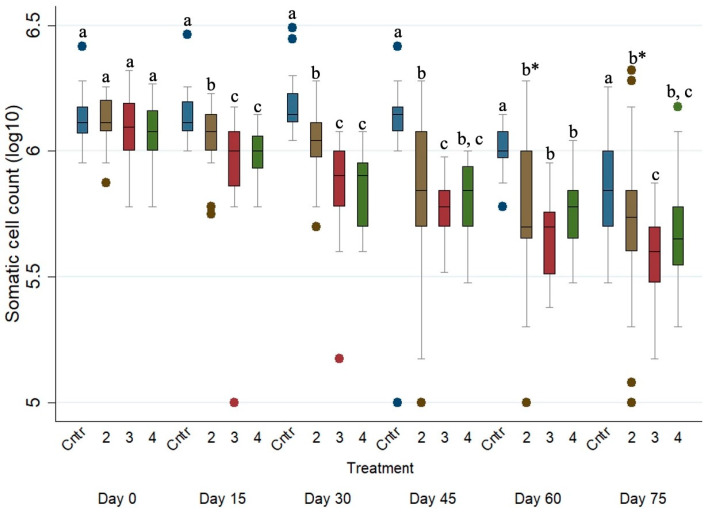
Box plots indicating the effects of the different anthelmintic treatments on somatic cell count (log_10_ SCC) among the 4 different groups per sampling occasion—Group 1: control group or untreated animals (blue color); Group 2: albendazole-treated animals (brown color); Group 3: pour-on eprinomectin-treated animals (red color); Group 4: injectable eprinomectin-treated animals (green color). a, b and c indicate statistically significant differences (*p* ≤ 0.05) among the 4 different treated groups per sampling; * indicates statistical differences at *p* < 0.001.

**Table 1 animals-15-01028-t001:** Proportions (%) of L_3_ larvae (per parasite taxon) per group of ewes in all studied regions at Days 0, 15, 30, 45, 60 and 75.

Group	Day	Nematodes’ Genera			
		Tel	Hae	Tri	Others
1	0	51.3 ^a^	42.7 ^a^	3.0 ^a^	3.0 ^a^
	15	51.0 ^a^	44.0 ^a^	2.0 ^a^	3.0 ^a^
	30	50.4 ^a^	42.6 ^a^	3.0 ^a^	4.0 ^a^
	45	52.7 ^a^	43.3 ^a^	2.0 ^a^	2.0 ^a^
	60	55.0 ^a^	39.0 ^a^	2.2 ^a^	3.8 ^a^
	75	50.0 ^a^	43.0 ^a^	3.0 ^a^	4.0 ^a^
2	0	54.0 ^a^	41.0 ^a^	3.2 ^a^	1.8 ^a^
	15	70.0 ^b^	27.0 ^b^	0.0 ^b^	3.0 ^a^
	30	77.8 ^b^	19.2 ^b^	0.0 ^b^	3.0 ^a^
	45	78.8 ^b^	18.2 ^b^	0.0 ^b^	3.0 ^a^
	60	70.5 ^b^	25.0 ^b^	2.0 ^a^	2.5 ^a^
	75	70.0 ^b^	25.0 ^b^	2.0 ^a^	3.0 ^a^
3	0	54.0 ^a^	41.0 ^a^	2.0 ^a^	3.0 ^a^
	15	54.4 ^a^	45.6 ^a^	0.0 ^b^	0.0 ^b^
	30	53.0 ^a^	47.0 ^a^	0.0 ^b^	0.0 ^b^
	45	58.6 ^a^	41.4 ^a^	0.0 ^b^	0.0 ^b^
	60	60.0 ^a^	33.5 ^a^	3.5 ^a^	3.0 ^a^
	75	60.0 ^a^	33.5 ^a^	3.5 ^a^	3.0 ^a^
4	0	55.4 ^a^	39.6 ^a^	1.8 ^a^	3.2 ^a^
	15	58.4 ^a^	41.6 ^a^	0.0 ^b^	0.0 ^b^
	30	55.0 ^a^	45.0 ^a^	0.0 ^b^	0.0 ^b^
	45	57.0 ^a^	38.0 ^a^	3.0 ^a^	2.0 ^a^
	60	53.0 ^a^	40.5 ^a^	3.5 ^a^	3.0 ^a^
	75	54.0 ^a^	39.5 ^a^	3.3 ^a^	3.2 ^a^

^a, b^ Different superscripts in the same column indicate significant differences in the proportions of L_3_ larvae (per parasite taxon) per sampling occasion among four groups of ewes (*p* ≤ 0.05). Tel: *Teladorsagia* spp., Hae: *Haemonchus* spp., Tri: *Trichostrongylus* spp. Group 1: No treatment (control group), Group 2: Albendazole *per os* treatment at Day 0, Group 3: Eprinomectin pour-on treatment at Day 0, Group 4: Eprinomectin injectable treatment at Day 0. Others: *Chabertia* spp. and *Bunostomum* spp.

**Table 2 animals-15-01028-t002:** Mean (±SD) fecal egg counts (epg) per group of ewes and the anthelmintic efficacy (%) of albendazole (Group 2), pour-on eprinomectin (Group 3) and injectable eprinomectin (Group 4) across the study.

Day	Group 1	Group 2		Group 3		Group 4	
	Fecal Egg Count (epg)	Fecal Egg Count (epg)	Efficacy (%)	Fecal Egg Count (epg)	Efficacy (%)	Fecal Egg Count (epg)	Efficacy (%)
15	1243.3 (±712.9)	149.3 (±124.1)	88.0	0.8 (±6.4)	99.9	1.7 (±9.1)	99.9
30	1528.3 (±719.4)	152.5 (±277.3)	90.0	28.3 (±63.1)	98.1	19.2 (±49.4)	98.7
45	1554.2 (±688.3)	251.5 (±303.1)	83.8	58.3 (±139.9)	96.2	50.8 (±113.9)	96.7
60	1613.3 (±680.3)	454.2 (±437.8)	71.8	73.3 (±145.7)	95.5	88.3 (±168.9)	94.5
75	1825.0 (±744.6)	565.0 (±248.4)	69.0	126.7 (±188.7)	93.1	143.3 (±198.6)	92.1

epg: eggs per gram of feces.

**Table 3 animals-15-01028-t003:** The effects of the farm, anthelmintic treatment and sampling on the fecal egg counts (epg) in the studied ewe population.

Parameter	Category Level	B	SE	*p*-Values	95% CI	
					Lower	Upper
Farm	1	534.583	24.595	0.000	486.376	582.790
	2	175.208	20.423	0.000	135.178	215.238
	3	638.333	26.761	0.000	585.881	690.784
	4	860.833	25.449	0.000	810.954	910.712
	5	505.000	25.666	0.000	454.695	555.304
	6	370.000	23.752	0.000	323.446	416.553
	7	238.750	18.052	0.000	203.367	274.132
	8	Ref.				
	9	50.750	17.696	0.004	16.065	85.434
	10	572.916	26.442	0.000	521.090	624.742
	11	221.666	17.457	0.000	187.450	255.882
	12	40.000	18.741	0.033	3.266	76.733
Treatment	1	Ref.				
	2	−1048.569	47.577	0.000	−1141.819	−955.319
	3	−1270.903	53.771	0.000	−1376.293	−1165.512
	4	−1248.125	51.878	0.000	−1349.806	−1146.444
Sampling	1	Ref.				
	2	−739.687	35.974	0.000	−810.195	−669.179
	3	−634.479	36.554	0.000	−706.125	−562.833
	4	−587.854	37.163	0.000	−660.693	−515.014
	5	−509.270	35.490	0.000	−578.830	−439.710
	6	−401.562	38.442	0.000	−476.907	−326.217
Intercept	Continuous	1607.792	64.154	0.000	1482.052	1733.532

CI: confidence interval; B: coefficient; SE: standard error; Ref.: reference category; epg: eggs per gram of feces.

**Table 4 animals-15-01028-t004:** The effects of the anthelmintic treatment and sampling on the milk yield (mL) in the studied ewe population.

Parameter	Category Level	B	SE	*p*-Values	95% CI	
					Lower	Upper
Treatment	1	Ref.				
	2	60.264	31.656	0.083	−1.780	122.308
	3	168.097	78.707	0.051	77.767	243.963
	4	241.202	37.557	0.000	167.590	314.814
Sampling	1	Ref.				
	2	−18.968	18.814	0.313	−55.844	17.906
	3	−168.171	23.886	0.000	−214.988	−121.354
	4	−494.397	33.059	0.000	−559.192	−429.601
	5	−822.534	35.884	0.000	−892.866	−752.201
	6	−1004.976	33.997	0.000	−1071.609	−938.342
Intercept	Continuous	1444.084	61.667	0.000	1323.219	1564.949

CI: confidence interval; B: coefficient; SE: standard error; Ref.: reference category.

**Table 5 animals-15-01028-t005:** The effects of anthelmintic treatment and sampling on the fat concentration (%) of the milk in the studies ewe population.

Parameter	Category Level	B	SE	*p*-Values	95% CI	
					Lower	Upper
Treatment	1	Ref.				
	2	0.193	0.022	0.000	0.148	0.238
	3	0.293	0.021	0.000	0.250	0.336
	4	0.465	0.030	0.000	0.405	0.526
Sampling	1	Ref.				
	2	−0.149	0.036	0.000	−0.221	−0.077
	3	−0.149	0.046	0.001	−0.241	−0.057
	4	0.135	0.046	0.003	0.044	0.226
	5	0.559	0.049	0.000	0.463	0.655
	6	0.778	0.048	0.000	0.683	0.873
Intercept	Continuous	6.364	0.063	0.000	6.240	6.489

CI: confidence interval; B: coefficient; SE: standard error; Ref.: reference category.

**Table 6 animals-15-01028-t006:** The effects of anthelmintic treatment and sampling on the protein concentration (%) of the milk in the studied ewe population.

Parameter	Category Level	B	SE	*p*-Values	95% CI	
					Lower	Upper
Treatment	1	Ref.				
	2	0.075	0.015	0.000	0.046	0.104
	3	0.191	0.028	0.000	0.135	0.248
	4	0.247	0.031	0.000	0.185	0.309
Sampling	1	Ref.				
	2	0.123	0.020	0.000	0.082	0.164
	3	0.369	0.013	0.000	0.343	0.396
	4	0.524	0.020	0.000	0.484	0.564
	5	0.560	0.023	0.000	0.513	0.606
	6	0.613	0.019	0.000	0.575	0.651
Intercept	Continuous	4.461	0.095	0.000	4.274	4.648

CI: confidence interval; B: coefficient; SE: standard error; Ref.: reference category.

**Table 7 animals-15-01028-t007:** The effects of the anthelmintic treatment and sampling on the somatic cell counts (*10^3^/mL) of the milk of the studied ewe population.

Parameter	Category Level	B	SE	*p*-Values	95% CI	
					Lower	Upper
Treatment	1	Ref.				
	2	−128.324	29.523	0.000	−186.188	−70.460
	3	−265.724	29.348	0.000	−323.246	−208.202
	4	−199.322	32.666	0.000	−263.347	−135.297
Sampling	1	Ref.				
	2	−69.516	17.786	0.000	−104.378	−34.655
	3	−149.816	23.865	0.000	−196.592	−103.040
	4	−298.419	27.718	0.000	−352.746	−244.092
	5	−459.532	32.882	0.000	−523.981	−395.084
	6	−615.680	36.470	0.000	−687.160	−544.200
Intercept	Continuous	2530.673	144.675	0.000	2247.115	2814.231

CI: confidence interval; B: coefficient; SE: standard error; Ref.: reference category.

## Data Availability

The datasets used and/or analyzed during the current study are available from the corresponding author upon reasonable request.

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
