# Peer review of "Comparing Short Versus Long Persistency of Anthelmintics: Impacts on Dairy Sheep Production"

_animals, 2025, doi:10.3390/ani15071028_

Round 1
Reviewer 1 Report
Comments and Suggestions for Authors
Dear authors,
Thank you for the opportunity to review such an interesting article, which deals with important topics in the areas of Veterinary Medicine and Veterinary Parasitology: the impact of gastrointestinal nematodes' infections in sheep health and production indexes, and the effectiveness of several anthelmintics at a field level.
The manuscript is very well written, with a good English spelling, and with very complete "Introduction" and "Material and Methods" sections, and results carefully described and discussed in light of the available literature in these research topics.
Please find attached all my improvement suggestions, which were inserted in the document using the "Comments" tool of Adobe Acrobat Reader.
Well done!
Best regards

Reviewer 2 Report
Comments and Suggestions for Authors
It's a good article with interesting data; I have some suggestions that can further enhance the article originality.
Introduction section: You have reported citing previous work checking the quantitative and qualitative parameters of milk in dairy ewes for eprinomectin. So, does your claim to the first report remain valid?
Data is represented in table and figure form, and causing repetition of data, you can move tables to the separate supplementary material file.
The discussion section needs some attention. Data is represented in both table and figure form, and causing repetition of data, you can move tables to the separate supplementary material file. The discussion section can be modified by adding more data supporting or suppressing your findings.
You mentioned the reduced efficacy of benzimidazole, but what could be the possible reasons for this reduced efficacy of benzimidazoles mentioned that too.
Based on your study, what future action/suggestion have you mentioned that too
